# Predicting activatory and inhibitory drug–target interactions based on structural compound representations and genetically perturbed transcriptomes

**Won-Yung Lee, Choong-Yeol Lee, Chang-Eop Kim** *

Department of Physiology, College of Korean Medicine, Gachon University, Seongnam, Republic of Korea

* eopchang@gachon.ac.kr

## Abstract

A computational approach to identifying drug–target interactions (DTIs) is a credible strategy for accelerating drug development and understanding the mechanisms of action of small molecules. However, current methods to predict DTIs have mainly focused on identifying simple interactions, requiring further experiments to understand mechanism of drug. Here, we propose AI-DTI, a novel method that predicts activatory and inhibitory DTIs by combining the mol2vec and genetically perturbed transcriptomes. We trained the model on large-scale DTIs with MoA and found that our model outperformed a previous model that predicted activatory and inhibitory DTIs. Data augmentation of target feature vectors enabled the model to predict DTIs for a wide druggable targets. Our method achieved substantial performance in an independent dataset where the target was unseen in the training set and a high-throughput screening dataset where positive and negative samples were explicitly defined. Also, our method successfully rediscovered approximately half of the DTIs for drugs used in the treatment of COVID-19. These results indicate that AI-DTI is a practically useful tool for guiding drug discovery processes and generating plausible hypotheses that can reveal unknown mechanisms of drug action.

## 1. Introduction

Identifying drug–target interactions (DTIs) is an essential step in drug discovery and repurposing. Proper understanding of DTIs can lead to fast optimization of small molecules derived from phenotypic screening and elucidation of the mechanism of action for experimental drugs [1]. However, identifying a candidate drug for a putative target by relying solely on *in vivo* and biochemical approaches often takes 2–3 years with tremendous economic costs [2, 3]. Computational approaches have emerged as an alternative strategy for reducing the workload and resources by efficiently identifying potential DTIs. This strategy has the potential to accelerate the drug development process by prioritizing candidate compounds for putative targets or *vice versa*.

Conventional methods for predicting DTIs can be broadly categorized into docking simulations and ligand-based approaches [4, 5]. However, their prediction is often unreliable when

information files or at: https://bitbucket.org/NNSM/ai_dti.

**Funding:** This research was supported by a grant from the Korea Health Technology R&D Project through the Korea Health Industry Development Institute (KHIDI), funded by the Ministry of Health & Welfare, Republic of Korea (grant number HF20C0087) awarded to CK, the National Research Foundation of Korea (NRF), funded by the Korean government (MSIT) (grant number NRF-2020R1A6A3A13075094) awarded to WL, and the Ministry of Food and Drug Safety in 2021 (grant 21173MFDS561) awarded to CK. The funders had no role in study design, data collection and analysis, decision to publish, or preparation of the manuscript.

**Competing interests:** The authors have declared that no competing interests exist.

the 3D structure of a protein or target is unavailable or when an insufficient number of ligands is known for the target, respectively [6]. Recently, chemogenomic approaches have emerged as an alternative enabling large-scale predictions by leveraging recent advances in network-based approaches or machine learning techniques [7–11]. For example, Yunan et al. proposed DTI-Net, a network-integrated pipeline that predicts DTIs by constructing a heterogeneous network using the information collected from various sources [12]. Other researchers have proposed deep learning-based methods, such as convolutional neural network, graph convolutional network (GCN), and natural language processing, to predict novel DTIs [13–15]. Despite their state-of-the-art performance, these models predict simple interactions without the mode of action, necessitating further experimental validation to fully understand the mechanisms of action of the drug.

Researchers thus attempted to develop a model that predicts DTIs by specifying the mode of action. Specifically, Sawada et al. proposed a model that predicts activatory and inhibitory DTIs by combining transcriptome profiles measured after compound treatment and genetic perturbation [16]. Although the method showed the possibility of predicting DTIs with modes of action using transcriptome data, it did not provide a satisfactory tool that could be applied for drug discovery. First, employing compound-induced transcriptome profiles as a representative vector of compound limited the range of predictable compounds significantly. Second, the number of predictable activatory and inhibitory targets was 74 and 755, respectively, which covered only a fraction of the druggable targets. Finally, the employed algorithms, called joint learning, could not learn nonlinear relationships between input vectors and DTI labels, and thus, the performance of DTI prediction was insufficient. Therefore, there is still a pressing need to develop a method with superior performance for predicting a wide range of activatory and inhibitory targets for novel compounds and natural products.

In this paper, we present AI-DTI, a new computational methodology for predicting activatory and inhibitory DTIs, by integrating a mol2vec method and genetically perturbed transcriptomes (Fig 1). Employing mol2vec enabled our method to expand the drug space to most compounds with 2D structures. In addition, by inferring the target vector representation based on the protein–protein interaction (PPI) network, the number of predictable targets was expanded to cover a majority of druggable targets. We compared the performance of various classifiers on the training data set and selected the optimized classifier with the best performance. The prediction capacity of our model was also evaluated on independent datasets with unseen DTI pairs, and high-throughput biological assay results. Finally, as a case study, we evaluated whether our method could be applied to the prediction of DTIs for the novel disease, coronavirus disease 2019 (COVID-19). All these results demonstrate that AI-DTI is a practically useful tool for predicting unknown activatory and inhibitory DTIs, which provide new insights into drug discovery and help in understanding modes of drug action.

## 2. Materials and methods

### 2.1. Employing mol2vec-based compound features

The vector representation of compounds was obtained using mol2vec [17], a word2vec-inspired model that learns the vector representations of molecular substructures. Mol2vec applies the word2vec algorithm to the corpus of compounds by considering compound substructures derived from the Morgan fingerprint as "words" and compounds as "sentences". The vector representations of molecular substructures are encoded to point in directions similar to those that are chemically related, and entire compound representations are obtained by summing the vectors of the individual substructures. Among the mol2vec versions, we implemented the skip-gram model with a window size of 10 and 300-dimensional embedding of

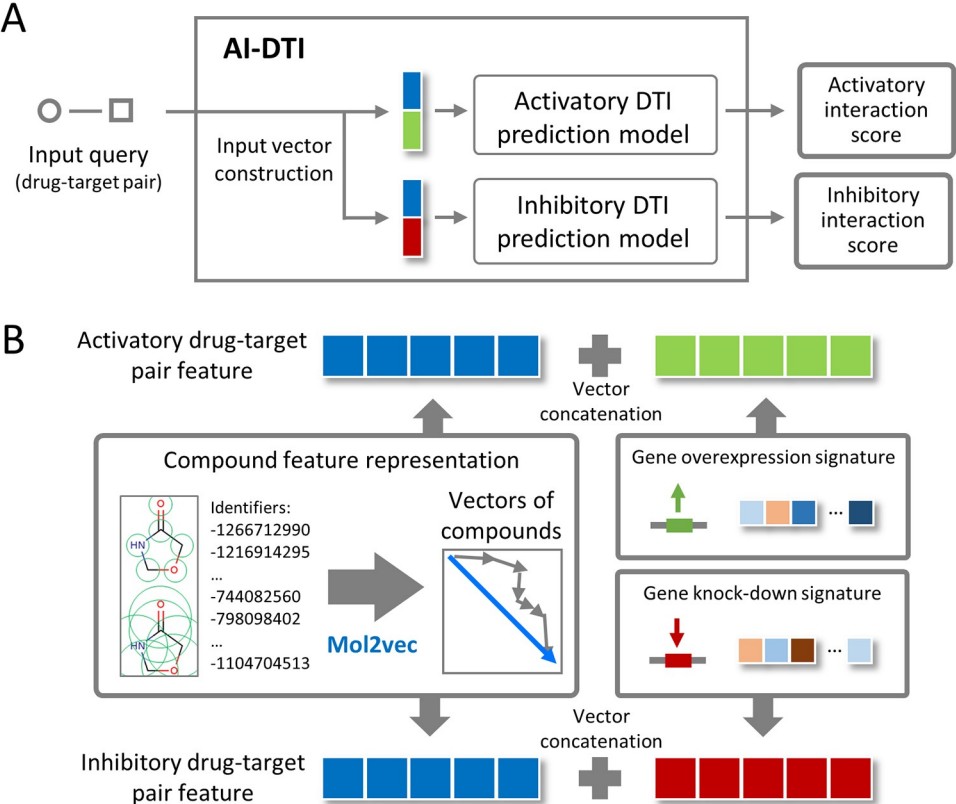

**Fig 1. Overview of the AI-DTI pipeline. (A)** A structure of AI-DTI. **(B)** Feature vector generation for activatory and inhibitory drug-target interactions (DTIs). For activatory DTIs, the feature vector was represented as a concatenation of the compound vector and aggregated gene overexpression signatures. For inhibitory DTIs, the feature vector was represented as a concatenation of the compound vector and aggregated gene knockdown signatures.

Morgan substructures, which demonstrated the best prediction capabilities in several compound property and bioactivity datasets.

## 2.2. Constructing genetically perturbed transcriptome-based target features

The genetically perturbed transcriptome of the L1000 dataset was downloaded from the Gene Expression Omnibus (accession number: GSE92742), which contains 473,647 signatures. Each signature (i.e., transcriptome profile) consists of a mediated z-score of 978 landmark genes whose expression levels were measured directly and 11,350 genes whose expression values were inferred from them. Landmark gene refers to one whose gene expression has been determined as being informative to characterize the transcriptome and which is measured directly in the L1000 assay. In our study, level 5 landmark gene data were used to represent the target vector. Level 5 data are a normalized dataset suggested by the LINCS team for use without additional processing. Among the types of perturbations, "cDNA for overexpression of wild-type gene" and "consensus signature from shRNAs for loss of function" were considered vector representations for activatory and inhibitory targets, respectively. From the downloaded data, the gene expression signatures of the landmark gene set were parsed using the cmapPy module [18], resulting in 36,720 gene knockdown and 22,205 gene overexpression signatures.

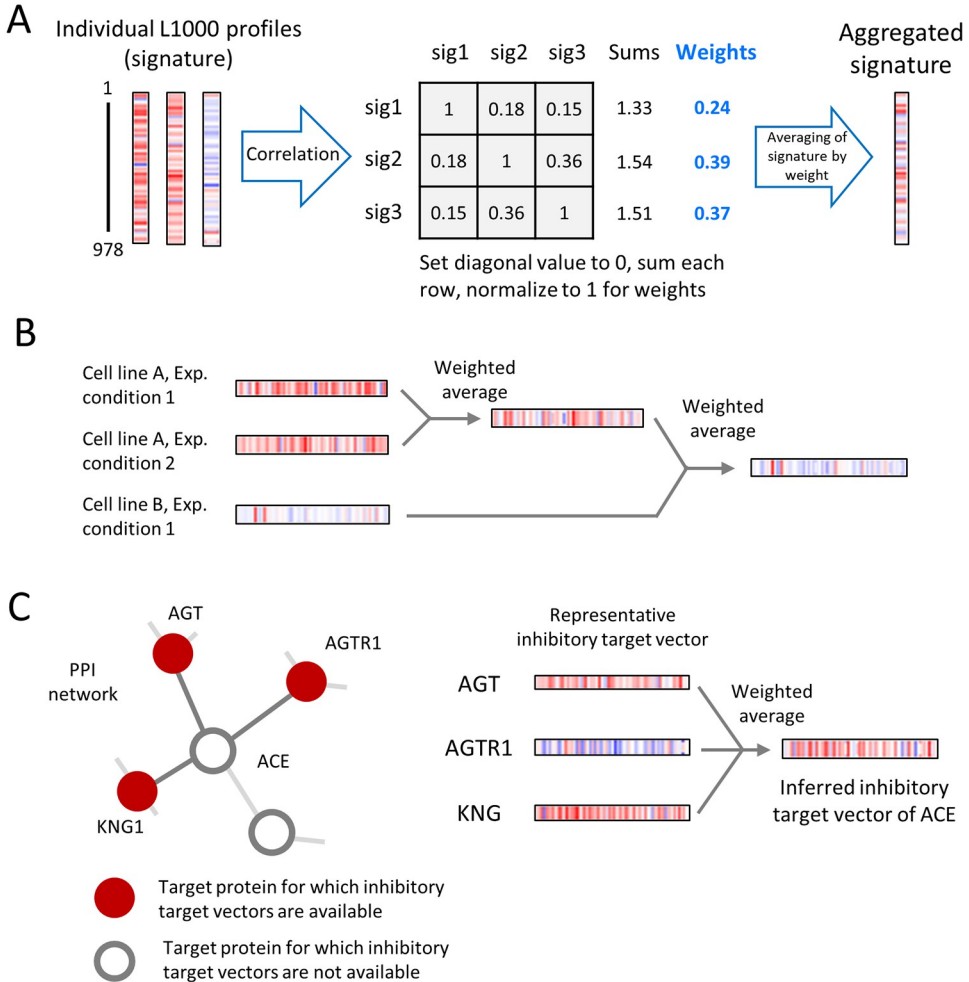

**Fig 2. Schematics of aggregation and inference for a genetically perturbed transcriptome. (A)** Weighted averaging for combining individual signatures into consensus gene signatures. Individual profiles were weighted by the sum of their correlations to other individual signatures and then averaged. **(B)** Generation of target vectors by cross-perturbation and cross-cell line aggregation. Multiple signatures measured after the same genetic perturbation in a specific cell, but with different perturbational doses or times, were first aggregated by weighted averaging. The same procedure was performed between multiple signatures measured after the same genetic perturbation in different cell lines. **(C)** Examples of inferring target vector representation. *AGT*, *AGTR1*, and *KNG1* interacted with *ACE* in the PPI network, and their inhibitory target vectors were available. The inhibitory target vector of *ACE* was inferred by aggregating these three representative inhibitory target vectors using weighted averaging.

The parsed data contained multiple gene expression profiles for single genetic perturbations measured in various cell lines and/or under perturbing conditions, which necessitated further preprocessing. To obtain the representative vector by each target, we applied a weighted average procedure divided into two steps: aggregation according to experimental conditions and aggregation across cell lines (Fig 2B). Before introducing the weighted average procedure, we described the process of applying the aggregation procedure. Specifically, the signatures measured after the same genetic perturbation in a specific cell, but with different perturbational dose or time, were first aggregated by weighted averaging. Then, the representative vector for a particular target is obtained by reapplying weighted averaging to these signatures (i.e., the aggregated signatures measured after the same genetic perturbation in different cell lines). This segmentation process reduces the potential biases on the representative vector calculation

that occurs when the number of genetically perturbed signatures skewed on a particular cell line. The process also allows us to compute embedding for targets that are at least 15% wider than when using gene expression in a single cell line (S1 Table).

The weighted average procedure is a method of weighted averaging of multiple signatures based on a pairwise correlation matrix (Fig 2A) [19, 20]. Suppose that $\mathbf{x}_t \in \mathbb{R}^{978}$ (t = 1, 2. . ., n) is a vector representing each L1000 signature of a landmark gene (normalized gene expression profiles for directly measured genes) for a specific functional perturbation, where t represents the elements of the signatures and n represents the total number of signatures to be aggregated. To generate the aggregated vector $\mathbf{x}_{Agg}$, a pairwise correlation matrix $\mathbf{R^{n \times n}}$ is defined as the Spearman coefficient between signature pairs,

$$\mathbf{R} = \begin{bmatrix} \rho_{11} & \cdots & \rho_{1n} \\ \vdots & \ddots & \vdots \\ \rho_{n1} & \cdots & \rho_{nn} \end{bmatrix} \qquad (1)$$

where $\rho_{ij}$ denotes the Spearman correlation coefficient between the signature pairs $\mathbf{x}_i$ and $\mathbf{x}_j$ for i, j $\in$ {1, 2, . . ., n}.

The weight vector ($\mathbf{w}$) is obtained by summing across the columns of $\mathbf{R}$ after excluding trivial self-correlation and then normalizing them,

$$\mathbf{w} = \frac{1}{\mathbf{j}^\mathrm{T}(\mathbf{R} - \mathbf{I})\mathbf{j}}(\mathbf{R} - \mathbf{I})\mathbf{j}, \qquad (2)$$

where $\mathbf{I}$ denotes the identity matrix, and $j$ denotes column vectors of 1s.

Finally, $\mathbf{x}_{Agg}$ is obtained from the average of $\mathbf{x}_t$ based on the weight vector $\mathbf{w}$,

$$\mathbf{x}_{Agg} = \sum\nolimits_{t=1}^{n} \mathrm{w}_t\mathbf{x}_t, \qquad (3)$$

where $\mathrm{w}_t$ denotes the t-th entries of $\mathbf{w}$. By aggregating the signatures across experimental conditions and cell lines, we obtained the representative vectors of 3,114 and 4,345 activatory and inhibitory targets, respectively, embedded in 978 dimensions (Fig 2B). The obtained target vector representation was used as target features of DTIs in an *original dataset* to be constructed later.

The target list of the obtained vector contained only a fraction of the druggable targets, thus significantly limiting the target space of our method. Therefore, the target space was extended by inferring the vector representation of activatory or inhibitory signatures based on the PPI network. The PPI network was constructed from the STRING database (v 11.0) [21] by setting the organism as "homo sapiens" and an interaction score > 0.9 (highest confidence score suggested by STRING). The vector representation of the activation or inhibition target was inferred by aggregating the vector representation of the interacting protein in the PPI network using a weight averaging procedure (Fig 2C). To ensure the quality of the data, we limited the inferred targets to proteins with at least three neighbours whose target vectors are available. The inferred target vector representation was used as target features of DTIs in an *additional dataset* to be constructed later.

## 2.3. Collection of activatory and inhibitory DTIs

Known activatory and inhibitory DTIs used as ground truth in our model were obtained from the Therapeutic Target Database (TTD) 2.0 (accessed October 15, 2020) [22] and DrugBank 5.1.7 (accessed January 12, 2021) [23]. We selected DTIs that explicitly defined activatory or

inhibitory interactions ("activator" or "agonist" for activatory DTIs and "inhibitor" or "antagonist" for inhibitory DTIs). Identifiers of compounds and targets with their annotations were standardized by PubChem chemical ID and gene symbols, respectively. The chemical structures of our dataset were retrieved in canonical SMILE format using the Python package PubChemPy. From TTD, we obtained 2,925 activatory and 32,417 inhibitory DTIs between 24,145 compounds and 2,117 targets. From DrugBank, we obtained 919 activatory and 4,719 inhibitory DTIs between 1,600 compounds and 1,022 targets.

## 2.4. Dataset construction

Two types of data sets were constructed respectively, which we refer to as the *original dataset* and the *additional dataset* (Fig 3A). Specifically, original dataset was constructed by selecting a known pair of activatory and inhibitory DTI pairs that include a compound for which ECFPs can be calculated and a target for which transcriptome data are available. Another dataset, *additional dataset*, was constructed by selecting DTI pairs that include a compound for which ECFPs can be calculated and a target for which inferred transcriptome data are available. Finally, the integrated dataset was constructed by combining these data sets and contained 1,755 activatory DTIs between 1,265 compounds and 273 targets, and 17,873 inhibitory DTIs between 12,259 compounds and 1,034 targets (Table 1).

Two independent datasets, Drugbank dataset and LIT-PCBA dataset, were constructed to measure the generalized ability of the trained model on predicting unseen DTIs (Fig 3B). We

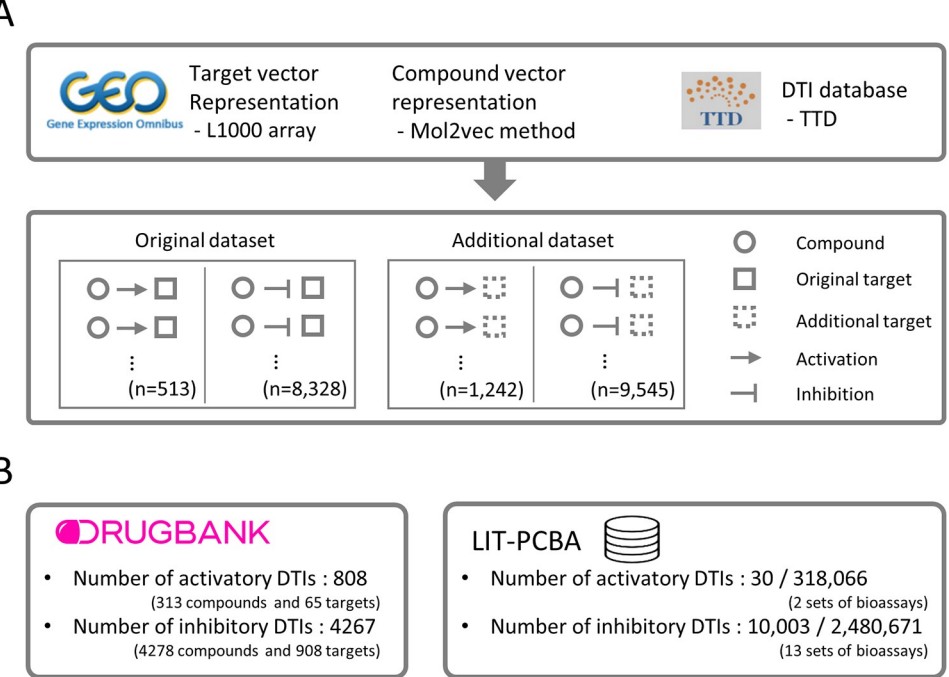

**Fig 3. Construction of a dataset of known DTIs and features. (A)** Training dataset construction. Transcriptome profiles were obtained from the L1000 array data and then aggregated to generate a representative target vector. A mol2vec method was used to generate representative vectors for compounds. DTIs with modes of action were collected from the TTD. The *original dataset* was constructed by selecting activatory and inhibitory DTI pairs that include a compound for which ECFPs can be calculated and an original target (i.e., a target for which genetically perturbed transcriptome data are available). The *additional dataset* was constructed by selecting activatory and inhibitory DTI pairs that include a compound for which ECFPs can be calculated and an additional target (i.e., a target for which inferred transcriptome data are available). **(B)** Independent dataset construction. Two independent datasets, Drugbank and LIT-PCBA datasets, were constructed to evaluate the reliability of predictions for unseen DTI in training datasets.

**Table 1. Overview of the drug–target interaction dataset for model training and external validation.**

| | Activatory DTIs | | | Inhibitory DTIs | | |
|---|---|---|---|---|---|---|
| | No. of compounds | No. of targets | No. of DTIs | No. of compounds | No. of targets | No. of DTIs |
| Internal sets | | | | | | |
| Original dataset | 457 | 87 | 513 | 6,789 | 702 | 8,328 |
| Additional dataset | 887 | 186 | 1,242 | 6,781 | 332 | 9,545 |
| Integrated dataset | 1,265 | 273 | 1,755 | 12,259 | 1,034 | 17,873 |
| External sets | | | | | | |
| DrugBank* | 374 | 172 | 808 | 1,217 | 909 | 4,267 |
| LIT-PCBA | 130,412 | 3 | 30/318,066# | 302,567 | 12 | 10,003/2,480,671# |

*A dataset with only DTIs unseen in the TTD dataset.

#Number of active and inactive DTIs.

selected two datasets as independent datasets because Drugbank provides broad and highly versatile DTI information, and LIT-PCBA offers systematically integrated high-throughput screening assay results. It is noteworthy that LIT-PCBA provides whether compounds are active/inactive for a specific target, so evaluation in this dataset can measure generalized performance in independent datasets where both negative samples and positive samples are explicitly defined. Each dataset was constructed by gathering a known pair of activatory and inhibitory pairs that include a compound with ECFPs and a target with (inferred) transcriptome data. Collectively, Drugbank datasets contained 808 activatory DTIs and 4,267 inhibitory DTIs (all positive samples), and LIT-PCBA datasets contained 318,096 activatory DTIs (30 positive and 318,066 negative samples) and 2,490,674 inhibitory DTIs (10,003 positive and 2,480,671 negative samples).

## 2.5. Training classifier models

The features of activatory and inhibitory DTIs were fed into classifier models to predict their interactions. The DTI prediction performance was evaluated by logistic regression (Logit), random forest (RF), multilayer perceptron (MLP), and cascade deep forest (CDF) models. RF is ensemble model that combine the probabilistic predictions of a number of decision tree-based classifiers to improve the generalization capability over a single estimator. MLP is a supervised learning algorithm that can learn nonlinear models. The architecture and hyperparameters of the MLP models in this study are summarized in S1 Fig. CDF employs a cascade structure, where the model consists of a multilayered architecture, and each level consists of an RF and extra trees [24]. Each level of cascade receives feature information processed by the preceding level and conveys its result to the next level. A key feature of the updated CDF model, deep forest 21, is that it automatically determines the number of cascade levels that are suitable for the training data by terminating the training procedure when performance improvements through adding cascade levels are no longer significant [25]. Greedy-search methods were used to select optimized hyperparameters of CDF as follows: the number of estimators in each cascade layer, the number of trees in the forest, the function to measure the quality of a split, maximum depth, minimum number of samples to be split, maximum features, and minimum impurity decrease. The detailed search range of the optimization is summarized in S2 Table.

## 2.6. Performance evaluation

The performance on the training set was evaluated using fivefold cross-validation (CV). In each fold, a training set was constructed by randomly selecting a subset of the 80% known DTI

pairs (assigned as the positive sample) and the randomly selected DTI pairs (assigned as the negative sample), and the test set was constructed by selecting the remaining 20% of the known DTI pairs and a matching number of randomly selected DTI pairs. For each fold of the predictive model, the following metrics were calculated:

$$\text{Precision} = \text{TP}/(\text{TP} + \text{FP})$$

$$\text{True positive rate} = \text{TP}/(\text{TP} + \text{FN})$$

$$\text{False positive rate} = \text{FP}/(\text{FP} + \text{TN}),$$

where TP is true positive, FP is false positive, FN is false negative, and TN is true negative. We plotted the receiver operating characteristic (ROC) curves based on different recall and false positive rates and precision-recall curves based on different precision and recall values under the conditions of different classification cut-off values. Area under the receiver operating characteristic curve (AUROC) and Area under the precision-recall curve (AUPR) were calculated over each fold, and their average values were recorded as measures of model performance. AUPR provides a better assessment in highly skewed datasets, whereas AUROC is prone to be an overoptimistic metric. Thus, we used AUPR as the key metric for model selection [26, 27].

To evaluate the performance in the specific threshold in the ROC curve, the enrichment in true actives at a constant x% false positive rate over random picking (EFx%) was calculated as follows:

$$\text{EFx\%} = \frac{TP_{x\%}/(TP_{x\%} + FP_{x\%})}{(TP_{x\%} + FN_{x\%})/(TP_{x\%} + TN_{x\%} + FP_{x\%} + TN_{x\%})}$$

Where $TP_{x\%}$, $FP_{x\%}$, $FN_{x\%}$ and $TN_{x\%}$ are the number of true positive, false positive, false negative and true negative samples at the threshold showing a false positive rate of x%, respectively. The EFx% value represents the enriched true positive rate compared to the expected value at the threshold representing a false positive rate of x%. For example, the EF1% value indicates the enrichment of the true positive rate compared to the chance level (0.01) at the threshold where the false positive rate is 0.01.

## 3. Results

### 3.1. Overview of AI-DTI

We developed AI-DTI for the *in silico* identification of activatory and inhibitory DTIs. AI-DTI is composed of two models that predict activatory DTIs and inhibitory DTIs (Fig 1A). When an input query (drug-target pair) is received, AI-DTI transforms it into activatory and inhibitory DTI feature vectors, and then estimated their interaction scores using each prediction model. The feature vector of DTIs was constructed by concatenating the compound vector calculated by mol2vec and genetically perturbed transcriptomes (gene-over expression signatures for activatory DTIs and gene-knock out signatures for inhibitory DTIs, Fig 1B). The mol2vec method transforms the structural information of a 2D compound into a continuous multidimensional vector. A genetically perturbed transcriptome reflects the response of the biological system following genetic perturbations, which is associated with the biological responses when drugs activate or inhibit specific gene targets [28–30].

The procedure for constructing AI-DTI and demonstrating its performance on independent dataset consisted largely of the following three steps below. First, we constructed three types of data sets–*Original dataset*, *additional dataset*, and *integrated dataset*–consisting of

DTI labels and their feature vectors. *Original* and *additional datasets* were constructed by selecting known pairs of activatory and inhibitory DTIs that contained targets for which transcriptome data could be measured or inferred, respectively. *Integrated Dataset* refers the sum of these two datasets. We assigned known DTIs for each mode of action constructed from each dataset as positive samples. Due to the lack of an adequate golden-standard negative set, negative samples are inevitably generated by random selection of non-interacting pairs from these drugs and targets in each dataset. Then, we trained various classifiers to discriminate positive (activatory or inhibitory) and negative DTIs on *Original dataset* and selected the optimized one with the best performance. The performance of the classifier was also evaluated in *additional* datasets and *integrated* datasets. Finally, the generalized performance of the optimized classifier trained on the *integrated* dataset was measured using independent datasets consisting of unseen DTIs [23, 31].

## 3.2. Selecting an optimized classifiers of AI-DTI

We first aimed to select the optimized classifier of the model on the *original dataset*. The dataset contained 1,755 and 17,873 activatory and inhibitory DTIs (assigned as the positive sample). Because the golden negative data set is not available, we randomly selected the nonpositive samples as many as the number of positive samples and assigned them as negative samples. We trained Logit, RF, MLP, and CDF models on the dataset and then evaluated the performance. The performance of the model was evaluated under a condition where 5-fold CV was repeatedly conducted 5 times with different data split. Our results showed that the CDF model yielded the highest AUROC and AUPR values in both situations when predicting activatory or inhibitory targets (Table 2). We subsequently tried to optimize the CDF model and found that the highest AUROC and AUPR values were obtained in all situations when the following hyperparameters were selected: '500' as the number of trees and '8' as the number of estimators (S2 Table). The optimized CDF model achieved AUROC and AUPR values of 0.880 and 0.899 for predicting activatory DTIs and 0.935 and 0.946 for predicting inhibitory DTIs, respectively.

Actually, DTI prediction in real world is an imbalanced classification problem where positive labels are sparse, so the performance measured on the dataset in which positive and negative samples are balanced does not fully reflect the situations in real drug discovery scenarios. To mimic the practical situation in which positive DTI is sparse, we also performed an additional CV test, in which the negative set in the test data contained ten times more negative samples than positive samples. With this experimental setup, the known DTI (i.e., positive samples) accounts for only 9% of the total data set, allowing a performance assessment closer to the situation of real drug discovery. Although the scores dropped when compared to the

**Table 2. Assessment of performance using the original datasets through fivefold cross-validation.**

| | Sample ratio = 1:1 (mean±S.D) | | | | Sample ratio = 1:10 (mean±S.D) | | | |
|---|---|---|---|---|---|---|---|---|
| | Activatory DTIs | Inhibitory DTIs | Activatory DTIs | Inhibitory DTIs | Activatory DTIs | Inhibitory DTIs | Activatory DTIs | Inhibitory DTIs |
| | AUROC | AUPR | AUROC | AUPR | AUROC | AUPR | AUROC | AUPR |
| Logit | 0.725±0.032 | 0.690±0.041 | 0.823±0.007 | 0.806±0.006 | 0.697±0.027 | 0.166±0.016 | 0.823±0.007 | 0.340±0.004 |
| RF | 0.868±0.022 | 0.890±0.021 | 0.921±0.004 | 0.932±0.003 | 0.858±0.026 | 0.559±0.054 | 0.923±0.004 | 0.729±0.010 |
| MLP | 0.841±0.029 | 0.851±0.035 | 0.920±0.004 | 0.918±0.006 | 0.835±0.020 | 0.379±0.038 | 0.916±0.004 | 0.559±0.018 |
| CDF[#] | **0.876±0.021** | **0.899±0.020** | **0.934±0.004** | **0.945±0.004** | **0.871±0.021** | **0.611±0.046** | **0.936±0.004** | **0.775±0.011** |

Boldface indicates the highest value for each performance metric. Logit, logistic regression; RF, random forest; MLP, multilayer perceptron; CDF, cascade deep forest.
[#] CDF model with 2 estimators in each cascade layer and 100 trees in each forest.

previous test, we observed that the optimized CDF and RF models still achieved high AUPR values (Table 2). The AUPR of the MLP model was significantly lower than that of the above two models, indicating that the performance of the MLP model was insufficient in the skewed dataset. Considering the highest performances in the experimental setup, we decided to employ the optimized CDF model as the classifier model of AI-DTI in subsequent analyses.

### 3.3. Performance comparison with previous models

The performances of our model were evaluated with previous approaches. We first focused on the performance evaluation in terms of different molecular embedding methods. Performance was measured under the same conditions as above in which 5-fold CV was repeated 5 times using a dataset in which the same number of positive and negative samples were selected. We found that the combination of mol2vec and CDF showed the highest performance for both AUROC and AUPR (Table 3). This result is consistent with a previous report that showed superior performance in the prediction of compound properties and bioactivity compared to Morgan fingerprints, chemical descriptors and some deep learning-based embedding models [17]. Taken together, we showed that mol2vec can provides rich information that can help accurately classify activatory and inhibitory DTIs.

The performance of our model was also compared with joint learning, a previous approach proposed by Sawada et al [16]. They constructed feature vectors based on the drug-induced signature and trained classifier models that predicts activatory and inhibitory DTIs for each target using joint learning. For comparison under the same conditions, we selected DTIs and their features from the *original dataset* for which drug-induced signatures were available in L1000 dataset. We obtained 55 activatory DTIs between 28 targets and 47 drugs and 592 inhibitory DTIs between 217 targets and 367 drugs. We note that any target included in all activatory DTIs have an insufficient number of DTIs (less than 5). Since this sparsity makes it difficult to properly assign a positive DTI to each fold during CV experiments, so we trained models and evaluate their performances focusing on the inhibitory DTI dataset which have sufficient number of DTIs for each target. As a result of comprehensive comparative evaluation over hyperparameters of joint learning and classifiers of AI-DTI, we found that AI-DTI showed higher AUROC and AUPR than joint learning (S3 Table). These results indicate that feature vectors calculated utilizing mol2vec could be more useful than drug-induced signatures in predicting DTIs.

### 3.4. AI-DTI can predict diverse druggable targets

The drawback of previous models using genetically perturbed transcriptomes is that the range of predictable targets is constrained to the targets for which the genetically perturbed

**Table 3. Assessment of performance across compound embedding methods.**

| | Activatory DTIs | | Inhibitory DTIs | |
|---|---|---|---|---|
| | **AUROC** | **AUPR** | **AUROC** | **AUPR** |
| MACCS | 0.883±0.023 | 0.852±0.028 | 0.941±0.005 | 0.923±0.006 |
| Morgan | 0.861±0.026 | 0.837±0.033 | 0.935±0.004 | 0.923±0.006 |
| Mol2vec | **0.885±0.024** | **0.863±0.027** | **0.945±0.004** | **0.934±0.005** |

Boldface indicates the highest value for each performance metric. Logit, logistic regression; RF, random forest; ERT, extremely randomized trees; MLP, multilayer perceptron; CDF, cascade deep forest.

# CDF model with 2 estimators in each cascade layer and 100 trees in each forest.

transcriptome is measured. For example, in a previous study [16], the number of predictable activatory and inhibitory DTI targets was only 77 and 769, respectively, covering only a fraction of druggable targets. To broaden the applicability of our method, we attempted to expand the target space of our model by inferring target vectors based on PPI networks (see *Materials and Methods*, Fig 2C). The assumption for using this method is that genetically perturbed transcriptomes are correlated with those of functionally interacting proteins. The inferring procedure calculates a representative vector for a target whose genetically perturbed transcriptome was not measured, thus enabling the construction of the input feature for wider targets. To check the reliability of the method, we first measured correlation between inferred data and genetically perturbed transcriptome. We estimated 1,673 activatory target vectors and 2,805 inhibitory target vectors for genes with measured transcriptome and computed the Spearman correlation between the inferred vectors and genetically perturbed transcriptome for the same gene. For comparison, we also calculated the Spearman correlations between genetically perturbed transcriptomes and inferred vectors of other genes. We found that the values of correlation between the same gene was significantly higher than those of other genes ($p < 0.001$ for both activatory and inhibitory targets, S2 Fig).

We then evaluated the predictive performance of the DTI in an *additional dataset* where target vectors consist of the inferred transcriptome. The performance of the model was measured in the same manner as in the above experiments. The results showed that the CDF model achieved satisfactory AUROC and AUPR values in the extended dataset (Table 4), indicating that activatory and inhibitory DTIs can be accurately predicted even with the inferred target vectors. We observed that there was no significant change in the performance when training the model by integrating the *original dataset* and extended dataset and when training the model using a separate dataset (Table 4). For ease of use, we decided to conducted subsequent analysis using a trained model in an integrated dataset that incorporates the *original dataset* and the *extended dataset*. It is noteworthy that our model trained in the datasets can predict more than 70% of druggable targets (targets that appeared in TTD), indicating that AI-DTI can be employed to predict a wide range of drug targets.

### 3.5. AI-DTI achieved substantial performance on independent datasets

To test the generalization abilities of the model, the performance of AI-DTI was further evaluated on independent datasets. We obtained activatory and inhibitory DTIs from DrugBank and selected DTIs that meets the following two criteria: (1) DTI pairs that include a compound for which ECFPs can be calculated and a target for which (inferred) transcriptome profiles are available and (2) DTIs that were not seen during the training phase (Table 1). We were

**Table 4. Assessment of the performance of the optimized CDF model for various datasets.**

| | Sample ratio = 1:1 (mean±S.D) | | | | Sample ratio = 1:10 (mean±S.D) | | | |
|---|---|---|---|---|---|---|---|---|
| | Activatory DTIs | | Inhibitory DTIs | | Activatory DTIs | | Inhibitory DTIs | |
| | AUROC | AUPR | AUROC | AUPR | AUROC | AUPR | AUROC | AUPR |
| Original dataset | 0.880±0.029 | 0.899±0.019 | 0.935±0.003 | 0.946±0.003 | 0.873±0.007 | 0.629±0.033 | 0.939±0.004 | 0.780±0.007 |
| Additional dataset | 0.873±0.011 | 0.869±0.013 | 0.953±0.002 | 0.957±0.002 | 0.864±0.012 | 0.430±0.030 | 0.955±0.002 | 0.800±0.008 |
| Separate model* | **0.875±0.011** | 0.878±0.011 | **0.944±0.002** | **0.952±0.002** | 0.867±0.009 | 0.488±0.022 | **0.947±0.002** | **0.790±0.004** |
| Integrated model# | **0.875±0.010** | **0.881±0.008** | 0.943±0.002 | 0.951±0.001 | **0.869±0.008** | **0.489±0.023** | 0.946±0.002 | 0.786±0.005 |

Boldface indicates the highest value for each performance metric between the separate model and integrative model.

*Models trained on the original and additional datasets separately.

#Models trained on an integrated dataset.

extremely careful that data leakage can significantly affect the performance of the model, and found that this process could successfully identify independent DTIs that are unseen in the training phase. All the remaining non-positive samples between drugs and targets were assigned to the negative samples. We predicted interaction scores using AI-DTI and evaluated their performances on predicting activatory and inhibitory DTIs. We found that our models achieved satisfactory AUROC and AUPR values (Fig 4A). Note that only a small ratio of the samples of the datasets are positive samples (1.26% and 0.03% for activatory and inhibitory DTIs, respectively), and this imbalance could be more unfavourable condition for DTI classification. Even on the highly skewed dataset, the optimized CDF-based model achieved the highest AUROC of 0.773 for activatory DTIs and 0.723 for inhibitory DTIs. The precision-recall curves also revealed that the performance of the optimized CDF model was still better than that of the other models. Taken together, these results indicate the generalizability of our model to predict DTIs in which the targets were unseen during the training phase.

Evaluating the performance on the dataset by assigning nonpositive samples as negative samples does not fully reflect practical drug discovery scenarios. To this end, we evaluated the performance of AI-DTI in another benchmark dataset, LIT-PCBA [31]. A key feature of LIT-PCBA is that it systematically integrates high throughput screening datasets consisting of components that active or inactive to targets. Therefore, performance evaluation on this dataset could reflect more realistic drug discovery scenarios where negative samples are explicitly defined as well as positive samples. We predicted activatory and inhibitory DTIs using the optimized CDF model and compared the performance with the baseline three virtual screening (VS) methods presented by LIT-PCBA, i.e., the 2D fingerprint similarity method, 3D shape similarity method, and molecular docking. To reduce the bias of the performance, we trained our methods ten times using different negative samples and measured the mean performance on a fully processed target set. The results show that AI-DTI achieved a higher mean AUROC than those achieved by conventional VS methods optimized with a max-pooling approach (Fig 4B). Specifically, our method achieved the highest AUROC values in all target sets (*FEN1*, *IDH1*, *KAT2A*, and *VDR*) where the other VS methods produced worse AUROC values than chance and for three target sets (*ALDH1*, *FEN1*, and *KAT2A*) that were unseen in the training phase. Moreover, our method achieved higher EF1% values than conventional methods for all activatory ligands (*ESR*_ago, *FEN1*, and *OPRK1*) and one inhibitory ligand (*FEN1*). It is worth noting that AI-DTI is a large-scale method that can predict a wide range of targets, whereas the other comparative models are local models built separately to predict specific protein targets. In summary, we found that our model still offers superior performance for classifying active and inactive compounds in high-throughput screening datasets containing DTIs with an unseen target and/or an unseen compound.

### 3.6. AI-DTI can predict DTIs for novel diseases

Another approach to assessing the practicality of a DTI predictive model could be whether it can aid the drug discovery process for unseen diseases. In other words, evaluating the performance of DTIs for unseen diseases is expected to assess their generalized ability to guide target-based drug discovery processes. We thus attempted to test whether AI-DTI could identify the DTIs of candidate drugs for COVID-19 treatment. Validated DTIs for COVID-19 were collected from DrugBank, and DTIs that met two criteria were further selected as follows: (1) DTIs containing a compound for which ECFPs could be calculated and a target with (inferred) transcriptome profiles available and (2) DTIs that were not seen during the training phase. We employed optimized CDF-based models to predict the activatory or inhibitory interaction scores for validated DTIs. We regard the validated DTI to be rediscovered when the predicted

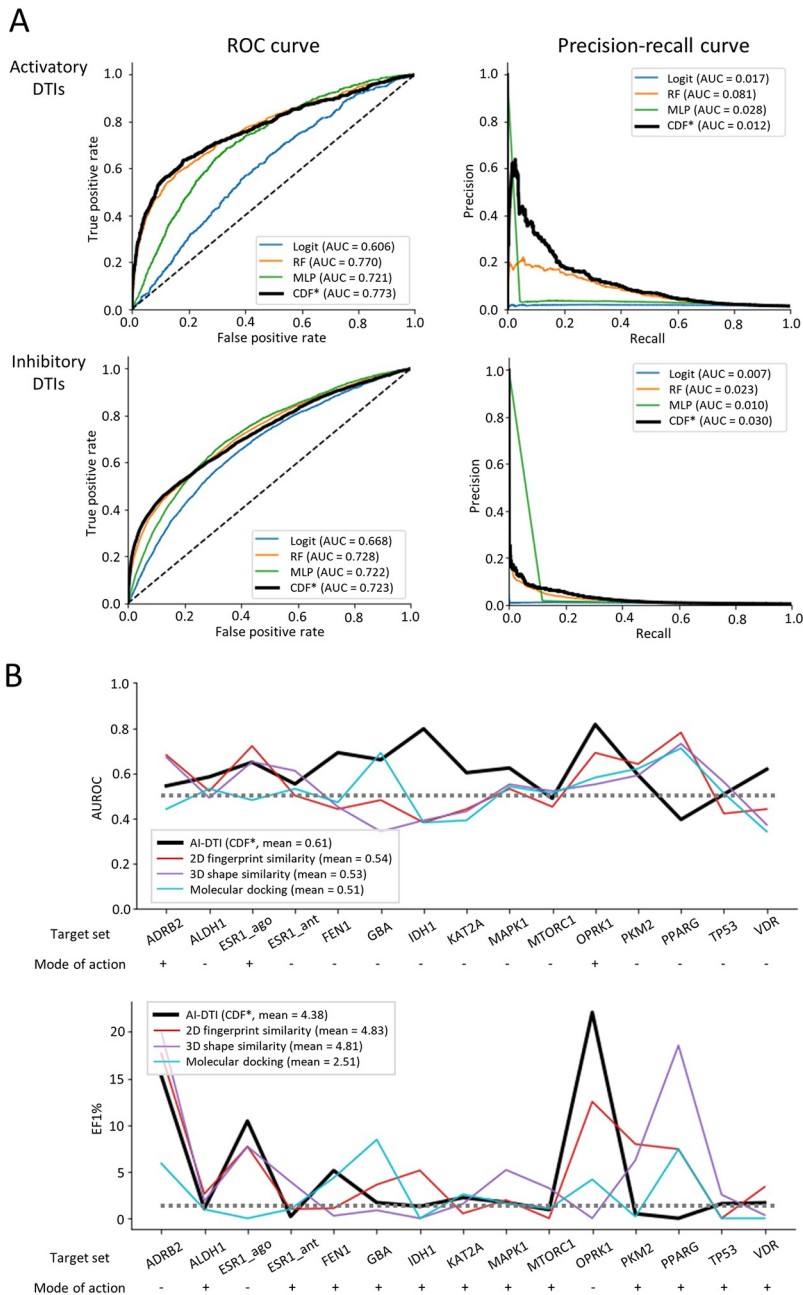

**Fig 4. Assessment of performance on independent datasets.** (A) Performance curves for activatory (top) and inhibitory (bottom) DTIs on the DrugBank dataset. *CDF model with 8 estimators in each cascade layer and 500 trees in each forest. Logit, logistic regression; RF, random forest; MLP, multilayer perceptron. CDF, cascade deep forest. **(B)** Performance comparison between our optimized model and the conventional virtual screening methods on LIT-PCBA. * CDF model with 8 estimators in each cascade layer and 500 trees in each forest.

score exceeds the default threshold of our model (0.5). To assess how uncommon the predicted score is, we constructed a reference distribution and compared a relative rank of the score (i.e., top %) to the reference distribution of interaction scores. The reference distribution was defined as the distribution of interaction scores calculated using our method for DTI pairs between 2500 FDA-approved drugs and a target of interest.

**Table 5. Predicted results for validated DTIs related to COVID-19.**

| Mode of action | Target name | Gene symbol | Drug name | Score* | Top percentage (rank)# |
|---|---|---|---|---|---|
| Activation | | | | | |
| | Peroxisome proliferator-activated receptor gamma | PPARG | Ibuprofen | **0.89** | 21.52 (538) |
| | Peroxisome proliferator-activated receptor alpha | PPARA | Ibuprofen | **0.61** | 41.12 (1028) |
| | Nuclear receptor subfamily 1 group I member 2 | NR1I2 | Dexamethasone | **0.75** | 5.12 (128) |
| | Annexin A1 | ANXA1 | Dexamethasone | 0.17 | 48.36 (1209) |
| | Glucocorticoid receptor | NR3C1 | Dexamethasone | **0.97** | 1.48 (37) |
| | Mu-type opioid receptor | OPRM1 | Metenkephalin | **0.92** | 1.16 (29) |
| | Delta-type opioid receptor | OPRD1 | Metenkephalin | **0.96** | 1.28 (32) |
| Inhibition | | | | | |
| | Tumour necrosis factor | TNF | Chloroquine | 0.21 | 86.84 (2171) |
| | Glutathione S-transferase A2 | GSTA2 | Chloroquine | 0.11 | 74.64 (1866) |
| | Glutathione S-transferase Mu 1 | GSTM1 | Chloroquine | 0.15 | 71.68 (1792) |
| | Toll-like receptor 9 | TLR9 | Chloroquine | 0.09 | 59.08 (1477) |
| | High mobility group protein B1 | HMGB1 | Chloroquine | 0.10 | 65.44 (1636) |
| | Tubulin beta chain | TUBB | Colchicine | 0.10 | 58.84 (1471) |
| | Prostaglandin G/H synthase 2 | PTGS2 | Ibuprofen | **0.95** | 2.84 (71) |
| | Cystic fibrosis transmembrane conductance regulator | CFTR | Ibuprofen | 0.19 | 80.56 (2014) |
| | Glutamate receptor ionotropic, NMDA 2B | GRIN2B | Ifenprodil | **1.00** | 0.08 (2) |
| | Glutamate receptor ionotropic, NMDA 1 | GRIN1 | Ifenprodil | **0.78** | 6.28 (157) |
| | G protein-activated inward rectifier potassium channel 1 | KCNJ3 | Ifenprodil | 0.07 | 93.24 (2331) |
| | G protein-activated inward rectifier potassium channel 4 | KCNJ5 | Ifenprodil | 0.07 | 93.24 (2331) |
| | G protein-activated inward rectifier potassium channel 2 | KCNJ6 | Ifenprodil | 0.07 | 93.24 (2331) |
| | Histone deacetylase 1 | HDAC1 | Fingolimod | **0.69** | 17 (425) |
| | Tyrosine-protein kinase JAK3 | JAK3 | Baricitinib | **0.60** | 2.44 (61) |
| | Tyrosine-protein kinase JAK1 | JAK1 | Baricitinib | 0.44 | 2.76 (69) |
| | Tyrosine-protein kinase JAK2 | JAK2 | Baricitinib | **0.65** | 4.2 (105) |
| | Protein-tyrosine kinase 2-beta | PTK2B | Baricitinib | 0.14 | 27.2 (680) |

* Scores that exceed the model's default threshold are in boldface.

# The top percentage and rank were calculated against the score for 2500 FDA-approved drugs and the corresponding genes.

We found that approximately half of the DTIs (12/25) were successfully rediscovered by our method, of which three and five activatory and inhibitory DTIs were found to be in the top 5%, respectively (Table 5). It is noteworthy that the targets of two activatory DTIs (metenkephalin—*OPRM1* and metenkephalin—*OPRM1*) and two inhibitory DTIs (ifenprodil—*GRIN1* and ifenprodil—*GRIN2B*) were included in the extended dataset, so the high top percentages of these DTIs support the reliability of our results within the extended target space. On the other hand, the low true positive rate of the inhibitory DTIs (33%, 6/18) might raise concerns about the reliability of our method's prediction results. However, except for the three targets (*TNF*, *HMGB1m* and *JAK1*), we found that all DTI scores between the FDA-approved drugs and targets showing false-negative results did not exceed the default threshold, which indicates that these false-negative results did not affect the precision of the predicted results. Also, we calculated a confusion matrix focused on DTIs between FDA-approved drugs and related targets with COVID-19, and found that AI-DTI achieved an F1-score more than twice the chance level (S3 Fig). To facilitate drug repurposing, we used our method to summarize a list of FDA-approved drugs yielding high scores for COVID-19 targets (S4 and S5 Tables).

## 4. Discussion

Accurately identifying DTIs with a mode of action is a crucial step in the drug development process and understanding the modes of action of drugs. Here, we present AI-DTI, a novel computational approach for identifying activatory and inhibitory targets for small molecules. By leveraging a mol2vec model and genetically perturbed transcriptome, AI-DTI was able to accurately predict active and inhibitory DTIs for a wide range of small molecule and drug targets. The comprehensive evaluation demonstrated that AI-DTI accurately predicts activatory and inhibitory DTI pairs, even in datasets containing sparse positive samples, DTI pairs unseen in the training phase, and high-throughput biological assay results. A case study of COVID-19 DTIs shows that AI-DTI can be used to prioritize activatory and inhibitory DTIs even for unseen diseases.

We believe that AI-DTI can bring significant contributions and advantages in drug discovery and research on the mechanisms of drugs. In drug discovery, our method can be applied to discover candidate compounds for diseases involving a variety of targets by providing large-scale predictions between a series of small molecules and a wide range of targets. Also, by predicting DTIs using only 2D structures, our method can generate plausible hypotheses for understanding the mechanisms of action including novel compounds and natural products whose known target information is scarce or sparse.

Among the employed classifier models, we found that the CDF model yielded the highest performances in our comprehensive experiments. Unlike the deep learning model, the CDF model automatically determines the complexity of the model in a data-dependent way with relatively few parameters and achieves excellent performance across various domains, including simple DTI prediction [7, 9, 25]. It is difficult to compare performance directly due to differences in datasets; however, the our method using the CDF model not only outperformed the previous model that predicts activatory and inhibitory DTIs but also competed with some state-of-the-art models that predict only simple interactions while requiring functional annotations of compounds such as drug-drug interactions, drug-disease relationships, and drug side effects [12, 32, 33].

We showed that AI-DTI is a practical tool that accurately predicts DTIs and their modes of action; however, there are several limitations of this study with potential for further improvement. First, the prediction performance can be further improved by applying advanced algorithms, such as GCN, which have been recently reported to show state-of-the-art performance [14]. Since the previous model still requires functional annotation of drugs, such as drug-drug interactions, an interesting future study will be to develop a model that predicts DTIs more accurately, even for novel compounds. Second, we used transcriptome profiles transduced with cDNA and shRNA as target vectors, which could include signal-to-noise issues due to background noise. The performance of our model may be improved further by upgrades based on large-scale datasets created using advanced techniques, such as CRISPR. A future direction of our work is to develop a versatile predictive model that accurately predicts DTIs with various modes of action.

## Supporting information

**S1 Table. Distribution of targets whose genetically perturbed transcriptome was measured by cell line.**
(XLSX)

**S2 Table. Search range and selected hyperparameter values for the cascade deep forest models.**
(XLSX)

**S3 Table. Performance comparison between joint learning and AI-DTI.**
(XLSX)

**S4 Table. Candidate FDA-approved drugs for COVID-19-related activatory targets.**
(XLSX)

**S5 Table. Candidate FDA-approved drugs for COVID-19-related inhibitory targets.**
(XLSX)

**S1 Fig. The architecture and hyperparameters of the MLP models.**
(TIF)

**S2 Fig. Distribution of spearman correlation coefficients between the inferred data and genetically perturbed transcriptome for the same gene (Within pair) and the other genes (Between pair).**
(TIF)

**S3 Fig. Predictive performance of AI-DTI on DTIs between FDA-approved drugs and COVID-19-related targets.**
(TIF)

## Author Contributions

**Conceptualization:** Won-Yung Lee, Choong-Yeol Lee.

**Data curation:** Won-Yung Lee.

**Formal analysis:** Won-Yung Lee.

**Funding acquisition:** Chang-Eop Kim.

**Investigation:** Won-Yung Lee.

**Methodology:** Won-Yung Lee, Chang-Eop Kim.

**Supervision:** Choong-Yeol Lee, Chang-Eop Kim.

**Validation:** Won-Yung Lee.

**Visualization:** Won-Yung Lee.

**Writing – original draft:** Won-Yung Lee.

**Writing – review & editing:** Choong-Yeol Lee, Chang-Eop Kim.

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
