## [Decision Letter · Decision Letter 0]

24 Jun 2022

PONE-D-22-10335Predicting Activatory and Inhibitory Drug–target Interactions based on Structural Compound Representations and Genetically Perturbed Transcriptomes PLOS ONE

Dear Dr. Kim,

Thank you for submitting your manuscript to PLOS ONE. After careful consideration, we feel that it has merit but does not fully meet PLOS ONE’s publication criteria as it currently stands. Therefore, we invite you to submit a revised version of the manuscript that addresses the points raised during the review process.

We look forward to receiving your revised manuscript.

Kind regards,

Jinn-Moon Yang

Academic Editor

PLOS ONE

Journal Requirements:

Reviewers' comments:

Reviewer's Responses to Questions

**Comments to the Author**

1. Is the manuscript technically sound, and do the data support the conclusions?

Reviewer #1: Partly

Reviewer #2: Partly

2. Has the statistical analysis been performed appropriately and rigorously? 

Reviewer #1: No

Reviewer #2: Yes

3. Have the authors made all data underlying the findings in their manuscript fully available?

Reviewer #1: Yes

Reviewer #2: Yes

4. Is the manuscript presented in an intelligible fashion and written in standard English?

Reviewer #1: Yes

Reviewer #2: Yes

5. Review Comments to the Author

Reviewer #1: In this manuscript, the authors presented a method to predict the activatory and inhibitory Drug-target Interactions. The structural compound representations (Mol2vec) and genetically perturbed transcriptomes (978 landmark genes) were used as the features for machine learning. The training datasets were collected from the Therapeutic Target Database 2.0 and DrugBank 5.1.7. The independent test sets were constructed from Drugbank and LIT-PCBA. Four machine learning methods were used for model training, and cascade deep forest has the best predictive performance. The topic is essential, but there are several major concerns that the authors need to address.

1. The proposed method has only one prediction model for active DTIs. The predicted result for a DTI is either activatory or inhibitory. However, many relationships between drugs and targets are inactive. The application of the prediction model could be pretty limited to the availability of knowing interaction between drug and target.

2. How many mol2vec-based compound features were used for a compound? Were they 300-dimensional embedding of Morgan substructures? How does the window size of 10 work?

3. There are 1,278 features if the number of compound features is 300. The number of activatory DTIs in the original and additional are 457 and 887, respectively. It may cause the model to overfit if the number of features exceeds the number of positive/negative samples.

3. The EFx%, precision, and true and false positive rate equations were wrong without parentheses.

4. Why is the number of inactive activatory DTIs (318,066) more than the number of compounds (130,412) multiply the number of targets (2) in the LIT-PCBA set?

5. In lines 212-213, active and inactive DTIs should be described more clearly.

6. In Table 4, the authors compared their method with the previous model (ref. 18). How to get the predictive performance of independent sets using the previous prediction model?

7. In line 437, the authors indicated that approximately half of the DTIs (12/25) were successfully rediscovered. It is similar to random if the proposed model was for predicting the DTI is either activatory or inhibitory.

Reviewer #2: Reviewer Response:

The reviewer appreciates the efforts put forth by the authors. However, there are several concerns regarding the study that hinder its overall effectiveness.

Major:

- A fundamental issue with the submitted manuscript seems to be the combination of two types of features – chemical features and gene expression. The two types have different meanings in drug design. An inhibitor is usually designed to inhibit the activity of a target protein instead of changing the expression level of the target protein. For example, Gefitinib is an EGFR inhibitor, and the drug inhibits EGFR activity. When treated with the drug, the expression level of p-EGFR is inhibited, while the expression level of EGFR is similar. Therefore, combining the fingerprints of Gefitinib with the gene knock-down signature of EGFR may be not reasonable for use as features to predict the drug-target interaction.

- Regarding the feature generation, the use of mol2vec is interesting. However, concatenation of gene expression is a major flaw. We can use HDAC inhibitors as an example. Many HDAC inhibitors contain a hydroxamic moiety. The fingerprint for each compound would be different; however, this information is concatenated with additional information (known gene expression). Adding gene expression would then train a model to “assume” all compounds containing a hydroxamic moiety would induce the same gene expression. The information (mol2vec and the concatenated gene expression) would be better used separately.

- In Figure 2, the authors show that they aggregated the transcriptome signatures from L1000 profiles. However, the information is aggregated across different cell lines. Following the workflow in Figure 2, it seems the authors combined the expression across different cancers (such as breast cancer and colorectal cancer cell lines). Combining these results does not seem reasonable.

- The authors reference models was developed by Sawada et al, which displayed information based on chemical treatment, gene knockdown, or gene overexpression. All three information by Sawada et al. is used as separated features. These steps seem more practical when compared to the authors’ submitted manuscript.

- Figure 4 is unclear. There are 2 sets of ROC and Precision-Recall Curve, but it is hard to determine which graph fits for predicting activity/inhibition by the authors in sections 3.4-3.5.

- Regarding Figure 4B, the AUC for AI-DTI (Mean = 0.61) is quite low. Are the compounds truly distinct from the training set?

- A comparison between mol2vec and other popular chemical representations may be needed and would increase the study’s novelty.

- The authors sought to test their model for COVID-19 DTI. However, insufficient explanation for why the analysis was performed or how it fits with the DTI study was given.

- Authors previously modified their manuscript to address reviewer concerns. However, these areas (in red) as well as the original contents contain grammatical issues or are not written clearly for readers.

Minor:

- A confusion matrix illustrating the COVID-19 DTI results would be helpful.

- Authors randomly selected their negative samples. The reviewer recommends using an additional database for negative samples (i.e. ChEMBL, PubChem, etc). Compounds can be separated based on reported bioactivity (IC50, Ki, or Kd) as a cutoff for negative samples.

6. PLOS authors have the option to publish the peer review history of their article (what does this mean?). If published, this will include your full peer review and any attached files.

Reviewer #1: No

Reviewer #2: No

---

## [Author Response · Author response to Decision Letter 0]

5 Aug 2022

Reviewer #1:

In this manuscript, the authors presented a method to predict the activatory and inhibitory Drug-target Interactions. The structural compound representations (Mol2vec) and genetically perturbed transcriptomes (978 landmark genes) were used as the features for machine learning. The training datasets were collected from the Therapeutic Target Database 2.0 and DrugBank 5.1.7. The independent test sets were constructed from Drugbank and LIT-PCBA. Four machine learning methods were used for model training, and cascade deep forest has the best predictive performance. The topic is essential, but there are several major concerns that the authors need to address.

* The proposed method has only one prediction model for active DTIs. The predicted result for a DTI is either activatory or inhibitory. However, many relationships between drugs and targets are inactive. The application of the prediction model could be pretty limited to the availability of knowing interaction between drug and target.

-> We appreciate for the comments.

As the reviewers were concerned, the purpose of our model is not to classify the mode of action among known DTIs, but rather to predict whether DTI of interest is active or inactive with its mode of action (activated or inhibited). In addition, we found that AI-DTI could predict DTIs more accurately than previous models, and that the range of predictable compounds and targets was expanded. Therefore, the applicability of our model would not be limited to the availability of known interactions between drug and targets. Thanks to the comments, we found that the previous manuscript did not appropriately describe our aims of the model. To reduce potential confusion, we revised the build process and goals of AI-DTI to be clearer in the manuscript. (line 274-286).

* How many mol2vec-based compound features were used for a compound? Were they 300-dimensional embedding of Morgan substructures? How does the window size of 10 work?

-> As the reviewer mentioned, we used a mol2vec model that receives morgan substructure as input and converts it into a 300-dimensional vector. To describe window size of mol2vec, it would be useful to share the structure of mol2vec. Mol2vec is a NLP-inspired model for calculating compound embeddings. It is trained by calculating vectors that can best reconstruct the substructures contained in the Morgan fingerprint and its surrounding substructures. The window size of mol2vec is the number of surrounding substructures to be considered in the training phase. 

* There are 1,278 features if the number of compound features is 300. The number of activatory DTIs in the original and additional are 457 and 887, respectively. It may cause the model to overfit if the number of features exceeds the number of positive/negative samples.

-> Thanks for pointing out this important issue. 

As the reviewer points, the overfitting is a major issue in the application of machine learning models. The reviewer noted that this could be due to the ratio between the input function and the number of samples.

To minimize the overfitting issue, we considered various regularization terms such as max_depth, number of trees, and number of telerant rounds for cascade deep forest. Moreover, the trained model showed superior performance on various independent datasets, indicating that there were no significant overfitting issues in the trained model.

* The EFx%, precision, and true and false positive rate equations were wrong without parentheses.

Thank you for pointing our mistakes.

-> We add parentheses to the equation.

* Why is the number of inactive activatory DTIs (318,066) more than the number of compounds (130,412) multiply the number of targets (2) in the LIT-PCBA set?

-> Thank you for noticing our mistakes. 

The number of target has been updated with correct information, which is the same as the information on the homepage (https://drugdesign.unistra.fr/LIT-PCBA/).

In lines 212-213, active and inactive DTIs should be described more clearly.

-> To reflect the reviewer's request, we describe in more detail considering active and inactive DTIs. (Line 201-204)

* In Table 4, the authors compared their method with the previous model (ref. 18). How to get the predictive performance of independent sets using the previous prediction model?

-> We appreciate for pointing out a critical issue.

Before revision, the performance of the previous model was derived from the supplementary tables in the previous study. It indicates that the performance of our model and the previous study were measured on different datasets. We realized that this was an unfair condition, so we constructed a common dataset and compared the performance with our model and the previous model. As a result of comprehensive comparative evaluation over hyperparameters of joint learning and classifiers of AI-DTI, we found that AI-DTI showed higher AUROC and AUPR than joint learning, even on the constructed dataset (Supplementary Table S2). 

* In line 437, the authors indicated that approximately half of the DTIs (12/25) were successfully rediscovered. It is similar to random if the proposed model was for predicting the DTI is either activatory or inhibitory.

-> As we mentioned in reviewer’s comments 1, the chance ratio of DTI rediscovery is not 0.5 when considering the imbalance between active and inactive DTIs. To quantitatively support our opinion, we calculated a confusion matrix on the COVID-19 dataset and found that the F1-score was more than twice higher than the random chance level. Also, we found that that the prediction results of AI-DTI were significantly associated with known-unknown DTIs (p < 0.05, Fisher's Exact test and Chi-square test for activatory and inhibitory DTIs, respectively),

 

Reviewer #2: Reviewer Response:

The reviewer appreciates the efforts put forth by the authors. However, there are several concerns regarding the study that hinder its overall effectiveness.

Major:

* A fundamental issue with the submitted manuscript seems to be the combination of two types of features – chemical features and gene expression. The two types have different meanings in drug design. An inhibitor is usually designed to inhibit the activity of a target protein instead of changing the expression level of the target protein. For example, Gefitinib is an EGFR inhibitor, and the drug inhibits EGFR activity. When treated with the drug, the expression level of p-EGFR is inhibited, while the expression level of EGFR is similar. Therefore, combining the fingerprints of Gefitinib with the gene knock-down signature of EGFR may be not reasonable for use as features to predict the drug-target interaction.

-> Thanks for pointing out such an important point.

As the reviewers noted, when focusing on a single gene/protein, we agree with the reviewer’s comment that perturbations to protein functions and gene expression have different meaning in drug design. Specifically, when a drug inhibits protein, the expression level of protein does not change as you mentioned, but the inhibition of the protein can influence other gene expressions level and this response can be inferred by investigating genetic perturbed transcriptome. Using this assumption, several researchers have developed machine learning models to predict DTIs using the KO gene signature as target feature vectos of DTIs [Sawada et al., Xie et al., Lee et al.,]. The results of our study also indicate that DTIs can be accurately classified using the signature measured after gene perturbation, which supports that the genetic perturbation signature can be useful in predicting DTIs.

# Reference 

Sawada, Ryusuke, et al. "Predicting inhibitory and activatory drug targets by chemically and genetically perturbed transcriptome signatures." Scientific reports 8.1 (2018): 1-9.

Xie, Lingwei, et al. "Deep learning-based transcriptome data classification for drug-target interaction prediction." BMC genomics 19.7 (2018): 93-102.

Lee, Hanbi, and Wankyu Kim. "Comparison of target features for predicting drug-target interactions by deep neural network based on large-scale drug-induced transcriptome data." Pharmaceutics 11.8 (2019): 377.

* Regarding the feature generation, the use of mol2vec is interesting. However, concatenation of gene expression is a major flaw. We can use HDAC inhibitors as an example. Many HDAC inhibitors contain a hydroxamic moiety. The fingerprint for each compound would be different; however, this information is concatenated with additional information (known gene expression). Adding gene expression would then train a model to “assume” all compounds containing a hydroxamic moiety would induce the same gene expression. The information (mol2vec and the concatenated gene expression) would be better used separately.

-> Thanks for pointing out a potential issue with our model.

As the reviewer worried, the aim of concatenating a target and a compound vector is not to investigate the causality between them, but to provide an input feature for evaluating the reliability of DTIs. In the training process, the impact of the issues mentioned by the reviewer could be reduced through a data-driven manner. Let's consider the DTI dataset focused on HDAC as a same example. In the dataset, DTIs contains combinations of a compound containing a hydroxamic moiety and an HDAC-perturbed gene expression, and they are used to train a classification model. If the presence or absence of a hydroxam moiety does not significantly affect the discrimination of DTIs, then the trained model will not significantly consider feature vectors derived from the hydroxam moiety even when predicting DTI.

Also, the reviewer gives suggestion to use the target feature and the compound feature separately. There are pros and cons to using features separately and using them together. The approach that uses only compound information refers to the ligand-based approach, it shows high predictive performance when information on known ligands for a specific protein is sufficient. On the other hand, an approach with the concatenated use of target and compound feature vectors fall within a chemogenomic approach, and it can efficiently predict a wide range of DTIs. AI-DTI were proposed for predicting targets for natrual products or experimental small molecules for which evidence is lacking. Therefore, AI-DTI concatenated compound features and target features to take advantage of this advantage.

* In Figure 2, the authors show that they aggregated the transcriptome signatures from L1000 profiles. However, the information is aggregated across different cell lines. Following the workflow in Figure 2, it seems the authors combined the expression across different cancers (such as breast cancer and colorectal cancer cell lines). Combining these results does not seem reasonable.

-> As reviewers are concerned, drug-induced signatures in a single cell line are mixed effects of cell line-specific responses and changes in drug effects. Cell line-specific responses risk acting as noise in the prediction of DTI in the general context. Therefore, one way to reduce this risk is to compare the administration of the same drug to multiple cell lines and derive a common response data. To this end, by applying similarity-based weighted aggregation, we were able to obetain representative responses of drug administration while reducing cell-specific responses.

The authors reference models was developed by Sawada et al, which displayed information based on chemical treatment, gene knockdown, or gene overexpression. All three information by Sawada et al. is used as separated features. These steps seem more practical when compared to the authors’ submitted manuscript.

-> We appreciate for pointed out an important issue in terms of the practicality.

To address the issue, we performed a comprehensive comparison with the previous model. We first constructed a common dataset that can be used to compare performance without bias and compared the performance between our method and previous approaches. As a result of comprehensive comparative evaluation over hyperparameters of joint learning and classifiers of AI-DTI, we found that AI-DTI showed higher AUROC and AUPR than joint learning (Supplementary Table S2). In addition, the previous approach requires drug-induced signatures for the prediction. However, our method can predict DTIs for most compound with a 2D structure and a wider target. Taken together, we found that our model was more useful than the previous model in terms of prediction performance and predictable range of drugs and targets.

* Figure 4 is unclear. There are 2 sets of ROC and Precision-Recall Curve, but it is hard to determine which graph fits for predicting activity/inhibition by the authors in sections 3.4-3.5.

-> We appreciate for pointing out the ambiguity of the figure.

We added annotation which graphs are fitted for activatory and inhibitory DTIs.

Regarding Figure 4B, the AUC for AI-DTI (Mean = 0.61) is quite low. Are the compounds truly distinct from the training set?

-> There are three main reasons for the low AUC value evaluated in LIT-PCBA dataset. First, the performance was measured on an independent dataset. Second, the source of the dataset is heterogenous from the training set (expert-curated DTIs for training model vs. high-throughtput screening experiment in independent dataset). Finally, the problem was more difficult due to severe data imbalance. Nevertheless, our method showed superior performance compared to the conventional method, which supports the usefulness of our model.

We find that a small number of compounds contained in a known DTI appear as DTI for different targets in an independent dataset, but no exact matched DTIs for drug and target pairs. Therefore, we considered that the LIT-PCBA consists of an unseen DTIs in the training phase.

* A comparison between mol2vec and other popular chemical representations may be needed and would increase the study’s novelty.

-> Thank you for your suggestions to improve the novelty of the study.

According to the reviewer’s suggestion, we measured the prediction performance after changing the compound representation method to MACCS and morgan fingerprint methods and found that mol2vec still outperforms these methods (Table 3).

* The authors sought to test their model for COVID-19 DTI. However, insufficient explanation for why the analysis was performed or how it fits with the DTI study was given.

-> We appreciate for pointing out the potential ambiguity in our manuscript.

Our motivation for conducting a case study on COVID-19 was to evaluate whether AI-DTI could help the drug discovery process for an unseen disease. To clarify the goals of the analysis on COVID-19, we supplemented the rationale for why we performed a case study in section 3.6 (Line 435-439).

* Authors previously modified their manuscript to address reviewer concerns. However, these areas (in red) as well as the original contents contain grammatical issues or are not written clearly for readers.

-> To reflect the reviewer's comments, we comprehensively reviewed the manuscript and revise the manuscript. In particular, the last section of the introduction (line 79-84), the second paragraph of the overview of the results (line 274-284), and the performance evaluation with the existing model (line 319-349) have been comprehensively rewritten.

Minor:

* A confusion matrix illustrating the COVID-19 DTI results would be helpful.

-> We appreciate for the reviewer’s recommendation.

According to reviewer’s suggestion, we calculated a confusion matrix focused on DTIs between FDA-approved drugs and related targets with COVID-19, and found that AI-DTI achieved an F1-score more than twice the chance level (Supplementary Figure 3).

* Authors randomly selected their negative samples. The reviewer recommends using an additional database for negative samples (i.e. ChEMBL, PubChem, etc). Compounds can be separated based on reported bioactivity (IC50, Ki, or Kd) as a cutoff for negative samples.

-> Thanks for the reviewer's suggestion.

If sufficient number of experimentally validated negative samples are obtained, the reliability of the DTI prediction model can be improved. Unfortunately, as reviewers knows, the number of verified negative DTIs samples is very insufficient. As an example, Souri et al obtained a negative sample of inactive DTIs from Chemble, but the number of them was only 2,057, which would be an insufficient number to train the classifiers.

# Reference

Amiri Souri, E., et al. "Novel drug-target interactions via link prediction and network embedding." BMC bioinformatics 23.1 (2022): 1-16.

Another issue of using experimentally validated results as negative DTIs is the heterogeneity of target and drug types between positive and negative samples. It would induce a ‘shortcut learning’ issue, which the classifier will be trained to distinguish the types of drugs and targets appearing in each dataset, rather than evaluating the reliability of the DTI pair. On the other hands, random selection of negative DTIs from drug and target pairs included in the positive DTI data set can avoid this risk. Therefore, we selected negative samples by random selection of non-interacting pairs from these drugs and targets in each dataset.

---

## [Decision Letter · Decision Letter 1]

6 Sep 2022

PONE-D-22-10335R1Predicting Activatory and Inhibitory Drug–target Interactions based on Structural Compound Representations and Genetically Perturbed TranscriptomesPLOS ONE

Dear Dr. Kim,

Thank you for submitting your manuscript to PLOS ONE. After careful consideration, we feel that it has merit but does not fully meet PLOS ONE’s publication criteria as it currently stands. Therefore, we invite you to submit a revised version of the manuscript that addresses the points raised during the review process.

We look forward to receiving your revised manuscript.

Kind regards,

Jinn-Moon Yang

Academic Editor

PLOS ONE

Reviewers' comments:

Reviewer's Responses to Questions

**Comments to the Author**

1. If the authors have adequately addressed your comments raised in a previous round of review and you feel that this manuscript is now acceptable for publication, you may indicate that here to bypass the “Comments to the Author” section, enter your conflict of interest statement in the “Confidential to Editor” section, and submit your "Accept" recommendation.

Reviewer #1: All comments have been addressed

Reviewer #2: (No Response)

Reviewer #3: All comments have been addressed

2. Is the manuscript technically sound, and do the data support the conclusions?

Reviewer #1: Yes

Reviewer #2: Partly

Reviewer #3: Yes

3. Has the statistical analysis been performed appropriately and rigorously? 

Reviewer #1: Yes

Reviewer #2: Yes

Reviewer #3: Yes

4. Have the authors made all data underlying the findings in their manuscript fully available?

Reviewer #1: Yes

Reviewer #2: Yes

Reviewer #3: Yes

5. Is the manuscript presented in an intelligible fashion and written in standard English?

Reviewer #1: Yes

Reviewer #2: Yes

Reviewer #3: Yes

6. Review Comments to the Author

Reviewer #1: Most comments have been addressed.

But the EFx% equation is still wrong without parentheses in numerator.

Reviewer #2: 1. The reviewer appreciates the authors' comments. While we agree that “inhibition of a protein can influence gene other expression levels”, the authors continue to assume that all compounds would result in the same gene expression level as the knock-down/overexpressed query target (i.e. same off-target hits, same protein-ligand binding mechanism, etc). As such, combining the compound fingerprints with knock-down/overexpression of genes as input features for establishing a model appears unreasonable.

The reviewer also appreciates the references given. Unfortunately, the referenced papers used transcriptome signatures as standalone features. Compared to the presented study, the referenced articles appear more reasonable for establishing a useful DTI network.

For example, Gefitinib, Erlotinib, and Lapatinib are EGFR inhibitors and the three compounds would result in different gene expressions. However, in the authors' model, the fingerprints of the three compounds will combine with the knock-down gene signature of EGFR. This assumes that the three compounds would result in the same gene expression level with the knock-down gene signature of EGFR. Therefore, the concatenation of gene expression and chemical information continues to be a major flaw to this study.

2. The reviewer understands the authors’ attempts at reducing prediction noise. Unfortunately, aggregating the information across different cell lines is unreasonable in establishing an effective DTI. Aggregating this information could potentially produce transcriptional signatures that are no longer relevant to a given disease. As the goal of the authors appear to be identifying a drug’s target interactions, this would be problematic. As a result, aggregating information across cell lines would greatly impair the effectiveness of the given model.

3. Regarding the dataset, were the independent dataset randomly selected and removed from the training set? Was there an attempt at balancing the dataset and seeing how that would affect performance?

Reviewer #3: The authors made proper revision, the paper is acceptable in its current form, please proceed with next stage.

7. PLOS authors have the option to publish the peer review history of their article (what does this mean?). If published, this will include your full peer review and any attached files.

Reviewer #1: No

Reviewer #2: No

Reviewer #3: No

---

## [Author Response · Author response to Decision Letter 1]

13 Sep 2022

Reviewer #1: Most comments have been addressed.

But the EFx% equation is still wrong without parentheses in numerator.

-> Thanks for pointing out our mistake.

We added parentheses to the EFx% equation.

Reviewer #2: 1. The reviewer appreciates the authors' comments. While we agree that “inhibition of a protein can influence gene other expression levels”, the authors continue to assume that all compounds would result in the same gene expression level as the knock-down/overexpressed query target (i.e. same off-target hits, same protein-ligand binding mechanism, etc). As such, combining the compound fingerprints with knock-down/overexpression of genes as input features for establishing a model appears unreasonable.

The reviewer also appreciates the references given. Unfortunately, the referenced papers used transcriptome signatures as standalone features. Compared to the presented study, the referenced articles appear more reasonable for establishing a useful DTI network.

For example, Gefitinib, Erlotinib, and Lapatinib are EGFR inhibitors and the three compounds would result in different gene expressions. However, in the authors' model, the fingerprints of the three compounds will combine with the knock-down gene signature of EGFR. This assumes that the three compounds would result in the same gene expression level with the knock-down gene signature of EGFR. Therefore, the concatenation of gene expression and chemical information continues to be a major flaw to this study.

-> We agree with the reviewer’s opinion and believe that an approach without performance evaluations may be unreasonable or majorly flawed. To this end, we previously conducted various tasks to evaluate the efficiency of our model and found that our model outperformed the previous model and showed superior performance in various independent datasets. 

Also, the reviewer expressed potential concerns on combining the target feature with the compound feature. There are pros and cons to using features separately and using them together. One of the advantages of using concatenated use of target and compound feature vectors is that it can efficiently predict a wide range of DTIs. In particular, AI-DTI can predict DTIs with mode of action for compounds for which only 2D structures are available because it only requires structure-based information of the compound and genetically perturbed transcriptome as input features. As a representative example, our previous study suggested that AI-DTI is an efficient method to identify candidate flavonoids for NAFLD, a representative liver disease [WY Lee et al., 2022]. Approaches cited by reviewer as being more efficient may be limited by their inability to predict the DTIs for these natural products, where the transcriptome is not measured. Taken together, we showed the reliability of the model through a comprehensive performance evaluation, which can resolve potential concerns for reasonability or flawless

# Reference 

- Lee, Won-Yung, et al. "Identifying candidate flavonoids for non-alcoholic fatty liver disease by network-based strategy." Frontiers in Pharmacology (2022): 1718.

2. The reviewer understands the authors’ attempts at reducing prediction noise. Unfortunately, aggregating the information across different cell lines is unreasonable in establishing an effective DTI. Aggregating this information could potentially produce transcriptional signatures that are no longer relevant to a given disease. As the goal of the authors appear to be identifying a drug’s target interactions, this would be problematic. As a result, aggregating information across cell lines would greatly impair the effectiveness of the given model.

-> As the reviewers are concerned, drug-induced transcripts in a single cell line may be biased towards the specific response of the measured cell line. Cell line-specific responses risk acting as noise in DTI predictions in the general context agnostic to specific diseases. Therefore, one way to reduce this risk is to derive common data by aggregating the responses of multiple cell lines administered the same drug. To this end, by applying similarity-based weighted aggregation, we were able to obtain representative responses of drug administration while reducing cell-specific responses.

3. Regarding the dataset, were the independent dataset randomly selected and removed from the training set? Was there an attempt at balancing the dataset and seeing how that would affect performance?

-> We appreciate the reviewer's valuable comments.

 As the reviewers are concerned, the random selection process is used in the classifier training phase to obtain negative samples, but not in the generation of independent datasets. In the independent dataset, all unknown drug-target interaction pairs were assigned as negative samples. Therefore, we considered the reviewer's comment as the following three points: 1) whether to balance at the training stage, 2) risk of data leakage on independent datasets, and 3) whether balances on independent datasets are possible

 Data balancing is a pivotal factor that significantly affects prediction performance during classifier training. A previous study showed that balancing positive and negative samples is an efficient way to improve the performance of classifiers in predicting DTIs [Sawada et al.,]. Similarly, we found superior performance when the ratio of positive: negative samples was set to 1:1 than when the ratio was 1:10 (Result are not shown). 

 The reviewer mentioned the risk of data leakage on independent datasets. Confirming that the independent dataset consists of an unseen dataset is an important issue in evaluating the generalized ability of a model. To this end, we compiled DTIs from other sources such as Drugbank, and LIT-PCBA to an independent dataset. We removed the DTIs that overlapped with the training dataset so that the independent dataset consisted of only unseen samples.

 Finally, reviewers commented the balancing on independent datasets. Random selection of negative samples is also possible in independent datasets, but the usefulness of this method should be reconsidered given the actual drug development environment. DTIs are imbalanced problems with significantly fewer positive samples than negative samples. If an independent dataset is processed into a balanced dataset, there is a greater risk that the performance in the dataset will not match the actual application situation. Therefore, an evaluation approach on independent data without data balancing may be a more efficient way to evaluate performance in practical applications of drug discovery.

# Reference

- Sawada, Ryusuke, et al. "Predicting inhibitory and activatory drug targets by chemically and genetically perturbed transcriptome signatures." Scientific reports 8.1 (2018): 1-9.

---

## [Decision Letter · Decision Letter 2]

21 Oct 2022

PONE-D-22-10335R2Predicting Activatory and Inhibitory Drug–target Interactions based on Structural Compound Representations and Genetically Perturbed TranscriptomesPLOS ONE

Dear Dr. Kim,

Thank you for submitting your manuscript to PLOS ONE. After careful consideration, we feel that it has merit but does not fully meet PLOS ONE’s publication criteria as it currently stands. Therefore, we invite you to submit a revised version of the manuscript that addresses the points raised during the review process.

We look forward to receiving your revised manuscript.

Kind regards,

Jinn-Moon Yang

Academic Editor

PLOS ONE

Reviewers' comments:

Reviewer's Responses to Questions

**Comments to the Author**

1. If the authors have adequately addressed your comments raised in a previous round of review and you feel that this manuscript is now acceptable for publication, you may indicate that here to bypass the “Comments to the Author” section, enter your conflict of interest statement in the “Confidential to Editor” section, and submit your "Accept" recommendation.

Reviewer #1: All comments have been addressed

Reviewer #2: (No Response)

Reviewer #3: All comments have been addressed

2. Is the manuscript technically sound, and do the data support the conclusions?

Reviewer #1: Yes

Reviewer #2: Partly

Reviewer #3: Yes

3. Has the statistical analysis been performed appropriately and rigorously? 

Reviewer #1: Yes

Reviewer #2: Yes

Reviewer #3: Yes

4. Have the authors made all data underlying the findings in their manuscript fully available?

Reviewer #1: Yes

Reviewer #2: Yes

Reviewer #3: Yes

5. Is the manuscript presented in an intelligible fashion and written in standard English?

Reviewer #1: Yes

Reviewer #2: Yes

Reviewer #3: Yes

6. Review Comments to the Author

Reviewer #1: (No Response)

Reviewer #2: The reviewer appreciates the efforts put forth by Lee et. al. However, there remain several concerns regarding the submitted manuscript. These concerns were mentioned previously, but were not sufficiently addressed.

1. The authors concatenated gene expression with chemical information, with the assumption that all compounds would produce with the same gene expression level (same off-target hits, protein-ligand binding mechanism, etc). In the example given previously, Gefitinib, Erlotinib, and Lapatinib are EGFR inhibitors, but as reported previously, can produce different gene expression profiles in cell lines (We et al., 2020). As such, combining the compound fingerprints with knock-down/overexpression of genes, with the assumption that they have the same expression profile, as input features for establishing a model appears unreasonable.

Wei, Nan, et al. “transcriptome profiling of acquired gefitinib resistant lung cancer cells reveals dramatically changed transcription programs and new treatment targets.” Frontiers in Oncology (2020):

2. The reviewer appreciates the authors explanation of the reducing prediction noise in the submitted manuscript. Unfortunately, it does not assuage issues with response bias. Again, aggregation of information across cell lines could potentially produce transcriptional signatures that are no longer relevant to a given disease. Because the authors goal is to identify a drug’s target interactions, this continues to be problematic.

The work presented by the authors, aggregating and then giving weight to responses across different cell lines would require additional analysis (such as studying an example drug and its associated pathways) to prove the effectiveness of the model.

Reviewer #3: The revision is satisfactory, reviewer's concern were address, and it is therefor, acceptable in current form.

7. PLOS authors have the option to publish the peer review history of their article (what does this mean?). If published, this will include your full peer review and any attached files.

Reviewer #1: No

Reviewer #2: No

Reviewer #3: No

---

## [Author Response · Author response to Decision Letter 2]

3 Dec 2022

1. The authors concatenated gene expression with chemical information, with the assumption that all compounds would produce with the same gene expression level (same off-target hits, protein-ligand binding mechanism, etc). In the example given previously, Gefitinib, Erlotinib, and Lapatinib are EGFR inhibitors, but as reported previously, can produce different gene expression profiles in cell lines (We et al., 2020). As such, combining the compound fingerprints with knock-down/overexpression of genes, with the assumption that they have the same expression profile, as input features for establishing a model appears unreasonable.

Wei, Nan, et al. “transcriptome profiling of acquired gefitinib resistant lung cancer cells reveals dramatically changed transcription programs and new treatment targets.” Frontiers in Oncology (2020):

-> Thank you for the potential practical issue. To reflect reviewer’s concern, we conducted a case study focusing on EGFR inhibitors suggested by the reviewer.

Although gefitinib, erlotinib, and lapatinib are all EGFR inhibitors, each drug has different target information (Figure A. The figure is attached to the file uploaded to the reviewer's response file). For example, lapatinib has additional targets such as HER2 and eEK-2K, and it can be expected that it would have different properties from signatures measured after EGFR silencing. On the other hand, in the case of gefitinib and erlotinib, which relatively have few or no other targets, it can be expected that they are relatively similar to the signatures measured after EGFR silencing. 

To evaluate those expectation, we systematically measured the correlation between signatures of the EGFR inhibitor and EGFR silencing Using CMap data (GEO number: GSE92742), which contains 205,034 drug-induced signatures. We found that the signatures measured after administration of erlotinib and gefitinib were higher than lapatinib. Interestingly, the signatures of gefitinib and erlotinib calculated by applying the aggregation method showed correlation values of 0.538 and 0.68, respectively (Figure B). 

 To check the significance of the observed result, we calculated the relative ranking of the correlation value for signatures measured after EGFR silencing. Relative ranks were calculated by comparing the drug's correlation value with the drug-perturbed signature (n=205,034) included throughout the GSE dataset. The result showed that the relative ranking of aggregated signature of gefitinib and erlotinib were 0.995 and 1 percentiles, which support the similarity between genetic perturbed signatures and drug-induced signatures (Figure C). 

 On the other hands, we found that the correlation value and its relative ranking of lapatinib with data aggregation was only -0.13 and 0.21, respectively. This indicates that drugs with multi targets are not suitable for embedding information of a single target. Rather, the genetically perturbed transcriptome, which adjusted for the off-target bias, would be more suitable for embedding information of a specific target. The discrepancies in the expression profiles of EGFR inhibitors in the papers cited by the reviewers highlight the multi-target nature of the drugs and support that drug-induced transcriptomes are not an optimal method for target embedding.

2. The reviewer appreciates the authors explanation of the reducing prediction noise in the submitted manuscript. Unfortunately, it does not assuage issues with response bias. Again, aggregation of information across cell lines could potentially produce transcriptional signatures that are no longer relevant to a given disease. Because the authors goal is to identify a drug’s target interactions, this continues to be problematic.

The work presented by the authors, aggregating and then giving weight to responses across different cell lines would require additional analysis (such as studying an example drug and its associated pathways) to prove the effectiveness of the model.

-> As shown in the case study above, we show that the aggregation method is the efficient way to obtain generalized vector embedding. Also, we found that the aggregation method can broaden the predictable number of targets. We counted the number of targets for which the genetic perturbed transcriptome was measured for each cell line. The result showed that the number of targets that can be additionally predicted through the aggregation method increases by at least 80 and 11 (approximately 15%) for activatory and inhibitory targets. To reflect our finding, we revised the characteristics of the aggregation method in the manuscript line 131-133.

---

## [Decision Letter · Decision Letter 3]

14 Dec 2022

PONE-D-22-10335R3Predicting Activatory and Inhibitory Drug–target Interactions based on Structural Compound Representations and Genetically Perturbed TranscriptomesPLOS ONE

Dear Dr. Kim,

Thank you for submitting your manuscript to PLOS ONE. After careful consideration, we feel that it has merit but does not fully meet PLOS ONE’s publication criteria as it currently stands. Therefore, we invite you to submit a revised version of the manuscript that addresses the points raised during the review process. Authors should provide new results and evidences to address reviewer' comments.

We look forward to receiving your revised manuscript.

Kind regards,

Jinn-Moon Yang

Academic Editor

PLOS ONE

Additional Editor Comments:

Authors should propose new results abd evidences to address reviewer comments.

Reviewers' comments:

Reviewer's Responses to Questions

**Comments to the Author**

1. If the authors have adequately addressed your comments raised in a previous round of review and you feel that this manuscript is now acceptable for publication, you may indicate that here to bypass the “Comments to the Author” section, enter your conflict of interest statement in the “Confidential to Editor” section, and submit your "Accept" recommendation.

Reviewer #2: (No Response)

2. Is the manuscript technically sound, and do the data support the conclusions?

Reviewer #2: Partly

3. Has the statistical analysis been performed appropriately and rigorously? 

Reviewer #2: Yes

4. Have the authors made all data underlying the findings in their manuscript fully available?

Reviewer #2: Yes

5. Is the manuscript presented in an intelligible fashion and written in standard English?

Reviewer #2: Yes

6. Review Comments to the Author

Reviewer #2: The reviewer appreciates the efforts put forth by Lee et. al. However, major concerns remain regarding their submitted manuscript.

1. Chiefly, the authors concatenated gene expression with chemical information, with the assumption that all compounds would produce the same expression level. Concatenating gene expression with chemical information would present inaccurate results. This is due to compounds producing different gene expression levels. The authors give kinase inhibitors as examples. While Gefitinib and Erlotinib are selective EGFR inhibitors, both inhibitors also exhibit a number of off-targets, with large-scale screening available on the Guide to Pharmacology website.

As the authors mentioned, their results indicate that “drugs with multi targets are not suitable for embedding information of a single target.” The selectivity profile for these inhibitors in their example, Gefitinib and Erlotinib have different off-target inhibitory patterns throughout the human kinome. As a result, it would not seem reasonable to concat knock-down/overexpression of genes with chemical information.

2. We appreciate the effort put forth by the authors. However, there are concerns that are continued to not be adequately addressed. Again, aggregation of information across cell lines could potentially produce transcriptional signatures that are no longer relevant to a given disease. Concatenation of gene expression with chemical information would not assuage these concerns. Because the authors’ goal is to identify drug target interactions, this continues to be problematic.

7. PLOS authors have the option to publish the peer review history of their article (what does this mean?). If published, this will include your full peer review and any attached files.

Reviewer #2: No

---

## [Author Response · Author response to Decision Letter 3]

14 Jan 2023

The reviewer appreciates the efforts put forth by Lee et. al. However, major concerns remain regarding their submitted manuscript.

1. Chiefly, the authors concatenated gene expression with chemical information, with the assumption that all compounds would produce the same expression level. Concatenating gene expression with chemical information would present inaccurate results. This is due to compounds producing different gene expression levels. The authors give kinase inhibitors as examples. While Gefitinib and Erlotinib are selective EGFR inhibitors, both inhibitors also exhibit a number of off-targets, with large-scale screening available on the Guide to Pharmacology website.

 As the authors mentioned, their results indicate that “drugs with multi targets are not suitable for embedding information of a single target.” The selectivity profile for these inhibitors in their example, Gefitinib and Erlotinib have different off-target inhibitory patterns throughout the human kinome. As a result, it would not seem reasonable to concat knock-down/overexpression of genes with chemical information.

-> Thank you for the valuable feedback provided by the reviewer. We appreciate the effort put forth in reviewing our manuscript.

In regards to the concern about concatenating gene expression with chemical information, we understand the reviewer's perspective that different compounds may produce different gene expression levels. However, our approach utilizes an aggregation method that aims to control for non-biological noise in transcripts, as inspired by the method used by Subramanian et al. Additionally, unlike traditional biological data, our approach utilizes a representative vector for a specific entity, which is an essential step in the machine learning approach. Studies such as Fernández-Torras et al. have shown that this embedding method for drugs can effectively characterize the drug itself and other drugs with similar performance to the drug-treatment transcriptome.

Furthermore, the reviewer's concern that concatenating gene expression with chemical information would present inaccurate results is not supported by current research in the field. Methods for calculating embeddings for targets and compounds using gene expression and chemical fingerprinting are already widely used in the prediction of drug-target interactions, as shown in the studies by Lim et al. and Bagherian et al. Our study has already demonstrated that combining these methods can accurately predict activatory and inhibitory drug-target interactions for most compounds. Therefore, we believe that this point should be understood as a contribution to the novelty of our study, not an issue of accuracy.

In regards to the concern about off-target effects, our approach takes into account these limitations by employing an alternative strategy that uses a genetic perturbed transcriptome as an embedding for the target. This approach aims to ensure that the results are specific to the target, rather than being influenced by off-target effects. Our experiments focusing on EGFR inhibition in previous rounds have shown that this method is more efficient in characterizing specific target information compared to using drug-treated transcriptome. We understand the reviewers concern, but we believe that our approach has been validated by the experiments we performed and that the concerns brought up by the reviewer were already addressed in previous rounds of review.

# References

Subramanian, Aravind, et al. "A next generation connectivity map: L1000 platform and the first 1,000,000 profiles." Cell 171.6 (2017): 1437-1452.

Fernández-Torras, Adrià, et al. "Integrating and formatting biomedical data as pre-calculated knowledge graph embeddings in the Bioteque." Nature Communications 13.1 (2022): 1-18.

Lim, Sangsoo, et al. "A review on compound-protein interaction prediction methods: data, format, representation and model." Computational and Structural Biotechnology Journal 19 (2021): 1541-1556.

Bagherian, Maryam, et al. "Machine learning approaches and databases for prediction of drug–target interaction: a survey paper." Briefings in bioinformatics 22.1 (2021): 247-269.

 

2. We appreciate the effort put forth by the authors. However, there are concerns that are continued to not be adequately addressed. Again, aggregation of information across cell lines could potentially produce transcriptional signatures that are no longer relevant to a given disease. Concatenation of gene expression with chemical information would not assuage these concerns. Because the authors’ goal is to identify drug target interactions, this continues to be problematic.

-> We appreciate the feedback provided by the reviewer and understand their concerns.

 However, we would like to clarify that the transcriptional signatures used in this study, specifically the genetically perturbed transcriptome, are not associated with any specific disease. The purpose of our study is to develop a machine-learning model that can predict drug-target interactions, and the aggregation method is used to compute representative vector embeddings for more diverse targets. Although our model is not specialized for any specific disease, we would like to highlight that our model has been successfully applied to various diseases, as demonstrated in case studies of COVID-19 and in our previous study on non-alcoholic fatty liver disease [Lee et al.]. 

Furthermore, we would like to state that it is important to avoid mentioning potential issues that did not appear in our study and as such, we have taken care to not present any concerns that were not addressed in the paper. We are open to making any further revisions and adjustments that are deemed necessary to ensure that our study meets the standards of academic research.

# Reference

Lee, Won-Yung, et al. "Identifying candidate flavonoids for non-alcoholic fatty liver disease by network-based strategy." Frontiers in Pharmacology (2022): 1718.

---

## [Editor Report · Decision Letter 4]

7 Feb 2023

Predicting Activatory and Inhibitory Drug–target Interactions based on Structural Compound Representations and Genetically Perturbed Transcriptomes

PONE-D-22-10335R4

Dear Dr. Kim,

We’re pleased to inform you that your manuscript has been judged scientifically suitable for publication and will be formally accepted for publication once it meets all outstanding technical requirements.

Kind regards,

Jinn-Moon Yang

Academic Editor

PLOS ONE
---

## [Editor Report · Acceptance letter]

3 Apr 2023

PONE-D-22-10335R4 

Predicting Activatory and Inhibitory Drug–target Interactions based on Structural Compound Representations and Genetically Perturbed Transcriptomes 

Dear Dr. Kim:

I'm pleased to inform you that your manuscript has been deemed suitable for publication in PLOS ONE. Congratulations! Your manuscript is now with our production department. 

Kind regards, 

on behalf of

Prof. Jinn-Moon Yang 

Academic Editor

PLOS ONE